# Application of New Efficient Hoveyda–Grubbs Catalysts Comprising an N→Ru Coordinate Bond in a Six-Membered Ring for the Synthesis of Natural Product-Like Cyclopenta[*b*]furo[2,3-*c*]pyrroles

**DOI:** 10.3390/molecules25225379

**Published:** 2020-11-17

**Authors:** Alexandra S. Antonova, Marina A. Vinokurova, Pavel A. Kumandin, Natalia L. Merkulova, Anna A. Sinelshchikova, Mikhail S. Grigoriev, Roman A. Novikov, Vladimir V. Kouznetsov, Kirill B. Polyanskii, Fedor I. Zubkov

**Affiliations:** 1Organic Chemistry Department, Faculty of Science, Peoples’ Friendship University of Russia (RUDN University), Miklukho-Maklaya St., 6, 117198 Moscow, Russia; alexandrasantonova@gmail.com (A.S.A.); marina.vin1999@yandex.ru (M.A.V.); pakumandin@gmail.com (P.A.K.); fraumerk@gmail.com (N.L.M.); 2A. N. Frumkin Institute of Physical Chemistry and Electrochemistry, Russian Academy of Sciences, Leninsky pr. 31, bld. 4, 119071 Moscow, Russia; asinelshchikova@gmail.com (A.A.S.); mickgrig@mail.ru (M.S.G.); 3V. A. Engelhardt Institute of Molecular Biology, Russian Academy of Sciences, Vavilov Street, 32, 119991 Moscow, Russia; novikovfff@bk.ru; 4Laboratorio de Química Orgánica y Biomolecular, CMN, Universidad Industrial de Santander, Parque Tecnológico Guatiguara, Km 2 vía refugio, Piedecuesta A.A. 681011, Colombia; kouznet@uis.edu.co

**Keywords:** furan, Hoveyda–Grubbs catalysts, nitrogen–ruthenium coordinate bond, ring-rearrangement metathesis, cyclopenta[*b*]furo[2,3-*c*]pyrroles, 3-allyl-3a,6-epoxyisoindoles, IMDAF reaction

## Abstract

The ring rearrangement metathesis (RRM) of a *trans*-*cis* diastereomer mixture of methyl 3-allyl-3a,6-epoxyisoindole-7-carboxylates derived from cheap, accessible and renewable furan-based precursors in the presence of a new class of Hoveyda–Grubbs-type catalysts, comprising an N→Ru coordinate bond in a six-membered ring, results in the difficult-to-obtain natural product-like cyclopenta[*b*]furo[2,3-*c*]pyrroles. In this process, only one diastereomer with a *trans*-arrangement of the 3-allyl fragment relative to the 3a,6-epoxy bridge enters into the rearrangement, while the *cis*-isomers polymerize almost completely under the same conditions. The tested catalysts are active in the temperature range from 60 to 120 °C at a concentration of 0.5 mol % and provide better yields of the target tricycles compared to the most popular commercially available second-generation Hoveyda–Grubbs catalyst. The diastereoselectivity of the intramolecular Diels–Alder reaction furan (IMDAF) reaction between starting 1-(furan-2-yl)but-3-en-1-amines and maleic anhydride, leading to 3a,6-epoxyisoindole-7-carboxylates, was studied as well.

## 1. Introduction

The discovery of metathesis reactions has allowed the noticeable expansion of possibilities for the transformation of different unsaturated substrates, including those aimed at obtaining complex naturally occurring compounds and pharmacologically meaningful molecules, which is reflected in a number of recent reviews [1,2,3,4,5,6,7,8]. In particular, the ring-rearrangement metathesis (RRM) of bi- and polycyclic alkenes has made it possible to obtain systems that are practically inaccessible by means of other synthetic pathways [9,10,11,12,13,14,15,16,17,18]. Most often, for the rearrangement of bridged alkenes, the commercially available Grubbs (G) and Hoveyda–Grubbs (HG) ruthenium catalysts are used, which permits the reliable production of scaffolds possessing a wide range of potential biological activity due to the characteristics of their carbon skeletons [19,20,21,22,23]. Representative examples of the above-mentioned metathesis reactions clearly demonstrate a variety of possible transformation routes that give the structural diversity of the available final products (Scheme 1). As can be seen, the direction of the reaction may depend not only on the structure of the substrate, but also on the reaction conditions selected and the chosen catalyst.

Moreover, among all the variety of metathesis products known to date, there is a sole successful example of the HG-catalyzed RRM of 3a,6-epoxyisondole **1** which has produced the previously unknown heterocyclic system of cyclopenta[*b*]furo[2,3-*c*]pyrrole **2**, but in modest yield (31%) [24] (Scheme 2).

Hence, further investigations of the influence of catalysts and the structure of unsaturated substrates on the selectivity of metathesis reactions remain an ongoing trend in modern organic synthesis. The selection of main starting materials for the suitable unsaturated substrates like **1** is also an important task. In this context, it should be noted that the starting material for **1** is furfural, which is a renewable chemical and one of the most available agricultural byproducts from biomass [25,26]. In regard to design of new catalysts for RRM, our group recently prepared a new class of effective ruthenium catalysts comprising an N→Ru coordinate bond in a six-membered ring (**Cat.1** and **Cat.2**) (Figure 1) and found that they revealed a high level of activity (up to 10^−2^ mol%) in standard models for the olefin metathesis (styrene, allylbenzene, diethyl diallylmalonate, diallyltosylamide, norbornen/hex-1-ene, etc.) [27,28].

According to the statements above and with the knowledge that there are no reports on the efficient synthetic methods for diversely polyfunctionalized cyclopenta[*b*]furo[2,3-*c*]pyrroles using easily available 3-allyl-3a,6-epoxyisoindoles derived from 1-(furan-2-yl)-N-arylmethanimines through the intramolecular Diels–Alder furan (IMDAF) reaction, our research was focused on: (i) Establishing the optimal conditions for the diastereoselective preparation of 3-allyl-3a,6-epoxyisoindoles, according to the variables: solvent, acid catalysts, reaction time and temperature; (ii) With the determined conditions in hand, preparing diverse 3a,6-epoxyisondol-7-carboxylic acids and their methyl esters; (iii) Corroborating the spatial structure of isomeric acids and their esters; (iv) Studying the catalytic activity of ruthenium catalysts **1** and **2** in the RRM of 3a,6-epoxyisoindole-7-carboxylates in comparison with HG-II catalyst; and (v) With the established conditions in hand, preparing the desired cyclopenta[*b*]furo[2,3-*c*]pyrroles. All this is in order to develop new, short and efficient synthesis of natural product-like scaffolds from renewable furan-based precursors.

## 2. Results and Discussion

The principal synthetic scheme for the required 3-allyl-3a,6-epoxyisoindole-7-carboxylic acids **5** preparation includes two key steps and is based on readily available starting materials. The condensation of aryl amines (or benzyl amines) and furfural gives the corresponding Schiff bases **3**, which, being treated with Grignard reagents (allyl magnesium bromide or methallyl magnesium chloride), give rise to homoallylamines **4** in multigram quantities (Scheme 3) [29,30,31,32].

The subsequent reaction of furfurylamines **4** with maleic anhydride easily provides the corresponding isoindolocarboxylic acids **5**, which were isolated in good overall yields. The IMDAF reaction is highly stereoselective [33,34]; in the course of the tandem N-acylation/[4+2] cycloaddition sequence, only *exo*-adducts form as a mixture of two diastereomers (*trans*-**5A** and *cis*-**5B**) based on the orientation of the 3-allyl (or 3-methallyl) substituent relative to the 3a,6-epoxy-bridge (Figure 2).

As we showed earlier [29,30,31,32], the synthesized 3-allyl-3a,6-epoxyisoindole-7-carboxylic acids **5** are excellent precursors for heterocyclization into the corresponding isoindoloquinolines and isoindolobenzazepines. In this process, the isomeric composition of adducts **5A/5B** had no strong influence on the yield and stereochemistry of the target heterocycles. In contrast, as was demonstrated in the first experiments in this study, only derivatives of the *trans*-isomer **5A** are capable of further transformation under RRM conditions. Thus, in the second stage of this investigation, we tried to resolve the problem of the diastereoselectivity of the IMDAF reaction.

Considering the fact that reversible Diels–Alder reactions are most influenced by the reaction temperature and the polarity of the solvent [35,36,37,38], we compared the isomeric composition of the most diverse acids **5a**,**b**,**g**,**i**,**m** formed under the conditions indicated in Table 1. The application of Lewis acids (10–25 mol % of AlEt_3_, BF_3_·OEt_2_, TiCl_4_) as catalysts in MeCN, PhH, PhMe, or CH_2_Cl_2_ turned out to be ineffective, probably due to their fast binding by a free carboxyl group of the products and due to the difficulties associated with the purification of the resulting reaction mixtures. Without catalyst addition, the pure target products precipitated from all tested solvents after the period of reaction time indicated in Table 1, wherein acetonitrile showed the worst results both at room and at elevated temperatures, apparently due to the partial dissolution of the target acids. As a result, it was found that according to ^1^H-NMR analysis, the *cis*-isomer **B**, inactive in the metathesis reaction, mainly forms at r.t in CH_2_Cl_2_ (average predominance 92%), while the desired *trans*-isomer **A** is mainly produced at −16 °C in the same solvent (average predominance 68%).

Decreasing the temperature to below −30 °C sharply slowed down the rate of interaction between maleic anhydride and secondary amines **4**; however, at the same time, analysis of the data in Table 1 shows that the yields of the isomer **5A/5B** mixtures obtained in CH_2_Cl_2_ are 1.5–2 times lower compared to the mixtures isolated from benzene or toluene. Besides, the last two solvents provide a slight predominance of the isomer **5A** at both r.t and heating. It is also important to note that the reaction time was the shortest in boiling toluene (~1 h); nevertheless, for a reasonable comparison of the results in Table 1, all reactions were carried out in boiling solvents for 3 h. Thus, for further experiments, we synthesized mixtures of the *trans*-**5A**/*cis*-**5B** isomers either in dichloromethane at −16 °C or in PhMe at 3 h reflux, as indicated in Table 2.

The third part of the work is related to the study of the catalytic activity of ruthenium catalysts **1** and **2** (Figure 1) under the RRM protocol. According to the previous paper [27], the *N*,*N*-dimethyl catalyst **2** is the most thermally stable, while the morpholine derivative catalyst **1** is the most active (as it has the longest coordination N→Ru bond ~2.32 Å).

With this in mind, only two of these complexes were selected for the present study. The direct introduction of acids **5** into the RRM was obviously impossible due to their poor solubility in common organic solvents and due to the presence of a free carboxylic group in their structure, which is destructive to Hoveyda–Grubbs catalysts. Thus, methyl esters **6** of corresponding acids **5** were prepared according to the classical procedure (Scheme 4, Table 2).

As the entries **a**, **h**, **k**, **n** in Table 2 reveal, the *trans/cis* isomeric ratio of acids **5** does not change noticeably during the esterification process, i.e., retro-DA reaction or epimerization do not proceed under selected conditions. Esters **6** obtained in this way are crystalline, easily isolated substances that are readily soluble in dichloromethane.

Elucidation of the spatial structure of isomeric acids similar to **5A/5B** and esters similar to **6A/6B** has been previously done in the literature more than once [29,30,31,32,37,39,40,41,42], while as a rule, 2D NOE NMR data have been used for evidence of their structures. Here we present X-ray structural information for a pair of isomeric methyl esters **6eA/6eB**, which confirms the preceding conclusions concerning the location of the substituent in the third position of the heterocyclic core (Figure 3).

Single crystals of diastereoisomers **6e** were grown from an ether solution, and single-crystal XRD analysis revealed that both isomers comprised 3a,6-epoxyisondolo-7-carboxylates with one molecule in an asymmetric unit. *Trans*-**6eA** isomer crystallizes in the triclinic crystal system P-1 space group, while the *cis*-**6eB** isomer crystallizes in the orthorhombic crystal system P2_1_2_1_2_1_ space group. The configuration of the tricyclic system is identical for both isomers (see Appendix A for bond lengths and angles, and Appendix A for the overlay of the two molecules). The substituents in the third position of the heterocyclic core are located in *trans* or *cis* positions that influence the distance between atoms, which are key for an NOE analysis (Table 3). The phenyl rings of 3a,6-epoxyisondolo-7-carboxylates are twisted at different angles with respect to the heterocycle due to the different orientations of the bulky substituent in the third position. In the case of *cis*-isomer **6eB**, the angle between the C_6_ plane of phenyl atoms and the C(1)N(2)C(3) plane is equal to 44°, while for *trans*-isomer **6eA**, this angle is 125°. The orientation of the methyl esters group differs only slightly for both isomers (Figure 3).

The indicated X-ray parameters (for **6eA** and **6eB**) are in good agreement with the data from 2D NOESY NMR experiments carried out for *trans*- and *cis*-isomers of compounds **5e**,**k**,**m**,**p** and **6e**,**k**,**m**,**p** (**A/B**) (Figure 4).

All these compounds have a similar configuration of the tricyclic skeleton and side groups for different substituents and gave the same cross-peaks in the NOESY spectra. Key NOE interactions are indicated in Figure 4. There are strong NOE interactions between H(4)–H(7a) and H(5)–H(7) for both diastereomers, indicating their identical endo-configurations for protons H(7) and H(7a). The difference lies in the side allyl/methallyl group, which is present in *trans*- or *cis*-form. There are strong resulting cross-peaks in the NOESY spectra between H(4)–H2C for *trans*-diastereomer A and H(4)–H(3) and H(3)–H(7a) for *cis*-diastereomer **B** (green arrows in Figure 4), which indicates their spatial proximity (<3 Å) and fully corresponds to the XRD data (Table 3). Cross-peaks between these protons but for other diastereomers (**B** and **A**, respectively)—between H(4)–H(3) and H(3)–H(7a) for *trans*-diastereomer **A** and H(4)–H2C for *cis*-diastereomer **B**—are absent in the NOESY spectra almost completely (crossed-out red arrows in Figure 4). The side allyl/methallyl group is conformationally quite mobile in solution, and NOE data for it cannot be appropriately compared with XRD distances.

^1^H and ^13^C-NMR spectra for series **5/6** of *trans*- and *cis*-diastereomers **A/B** have very few characteristic signals by which they could be easily distinguished. However, they have several characteristic patterns of correlation peaks in 2D HSQC spectra (Figure 5 and see the ESI with assignments data), for example, for CH(4) and CH(5) alkene protons, *ortho*-CH/*meta*-CH for *para*-substituted aromatics at N(2), and aliphatic CH_2_-groups in allyl/methallyl fragment. In the whole, the absolute values of chemical shifts are non-characteristic, since there is a fairly large scatter of chemical shifts depending on the substituents. However, for a number of protons and carbons, the sequence of the signals and difference of their chemical shifts are characteristic and the same for all compounds. The H(3), H(4), H(7a), CH_2_ protons, and the CH(3), C(3a), CH_2_ carbons have such features. Data on them are given in Appendix A (see the ESI).

Therefore, the combination of the single-crystal XRD and NMR data allowed us to make an unambiguous assignment of isomers of all compounds **5** and **6** to the **A** or **B** series.

Considering that we were unable to achieve a noticeable predominance of the desired acids **5A** by varying the IMDAF reaction conditions (see Table 1), we paid close attention to finding preparative methods for the separation of isomeric esters **6A/6B**. We failed to find a suitable chromatographic system for the separation of isomers **6A/6B** on silica gel due to the closeness of their retention factors. Mixtures of diastereomers **6aA/6aB** (79/21), **6eA/6eB** (16/84), **6fA/6fB** (29/71) and **6pA/6pB** (57/43) were used as model compounds, and mixtures of EtOAc/hexane, MeOH/THF and CHCl_3_/MeOH were applied as eluents. However, our attempts to separate the same esters by fractional crystallization were more fruitful. For this purpose, fine powders of the stereoisomer mixtures were stirred for a half-hour at 45–50 °C in a methanol solution; then, the least soluble isomer **6B** was filtered off and the mother liquor was concentrated. After this treatment, the purity of individual isomers **6A** and **6B** exceeded 92% (diastereomeric excess (*de*) > 84%) in total yields of more than 94%. Repeating this procedure one more time resulted in virtually pure isomers. According to the described protocol, the diastereomers of 3-allyl substituted esters **6aA/6aB**, **6fA/6fB** and **6eA/6eB** were separated and tested as metathesis objects (Scheme 5), described in the third part of this manuscript. Separation of 3-methallyl substituted esters **6pA/6pB** by this way was less efficient, resulting in the formation of a 65/35 mixture of isomers after one “recrystallization” from MeOH.

The first experiments showed that solutions of the individual *cis*-isomers **6aB**, **6eB** and **6fB** in CH_2_Cl_2_ at both room and elevated temperatures underwent polymerization with catalysts **1**, **2** or HG-II (0.5–5 mol%). We were unable to isolate individual substances from metathesis product mixtures obtained under either an argon or an ethylene atmosphere (2 bar), except in two cases **6eB** (R^1^ = H, R^2^ = 4-ClC_6_H_4_) and **6fB** (R^1^ = H, R^2^ = 4-BrC_6_H). In these tests under an argon atmosphere, divinyl furo[2,3-*c*]pyrroles **8**, probably the products of a series of the sequenced intermolecular cross metathesis reactions, were localized in a very low yield (Scheme 6). The formation of this product only occurs if the reaction is carried out for 15 min or less; upon longer heating, products **8** are completely transformed into a polymer.

In these two cases, it turned out to be possible to isolate products using column chromatography. The spatial structure of one of them (**8b**) was entirely confirmed by X-ray analysis (Figure 6). Compounds **8a** and **8b** also can be obtained by reaction of the esters **6e** or **6f** with ethylene (2 bar, 100 °C, 6 h) in dichloromethane in presence of catalyst **2** in 5–8% yields.

However, it was found that the pure *trans*-isomers **6aA**, **6eA** and **6pA** are able to form tricyclic systems of cyclopenta[*b*]furo[2,3-*c*]pyrrole **7** in the RRM reaction (the last column of Table 2 and Table 4). Moreover, in contrast to the communications [9,10,11,17,19,21,22], in our case, the ring-rearrangement metathesis does not require the supporting application of ethylene that facilitates practical synthesis.

To achieve the maximum yield of the target tricycles **7**, the metathesis reaction was optimized in terms of solvent, temperature, duration and amount of catalyst. Due to the poor solubility of esters **6eA** (3-allyl) and **6pA** (3-methallyl) in benzene and *o*-dichlorobenzene, we chose not to use aromatic solvents; similarly, polar solvents such as THF or MeCN proved to be ineffective in view of their capacity to interact with the Hoveyda–Grubbs type catalysts at elevated temperatures. The best of the tested mediums turned out to be halogenomethanes (CH_2_Cl_2_ and CHCl_3_), and hence they were used as solvents in all further experiments.

None of the selected catalysts exhibited activity at temperatures below 50 °C, and no rearrangement products **7e**,**p** were fixed. Being the most active, the morpholine containing **Cat. 1** started to work at the temperature of boiling chloroform (~60 °C), causing the rearrangement of the test 3-allyl isoindolone **6e**. The more sterically demanding methallylic substrate **6pA** are not recyclized under these conditions. An increase in temperature up to 100 °C under microwave irradiation led to a rapid decomposition of **Cat. 1** (at 100 °C, the emerald-green color of the chloroform solution of **Cat. 1** turned brown almost immediately), and accordingly to low yields of the product **7p**. Obviously, higher temperatures and a more thermally stable catalyst should be used for a successful cyclization of methallyl derivatives **6i**–**p**.

N,N-Dimethylamino derivative (**Cat. 2**), with a shorter bond length between nitrogen and ruthenium (the N→Ru bond is 2.24 Å [27]), is stable at least up to 140 °C [27], and therefore turned out to be effective for the preparation of cyclopenta[*b*]furo[2,3-*c*]pyrrole **7p** at temperatures above 100 °C in microwave-assisted experiments (entries 17–26, Table 4). It should be noted that the amount of catalysts **1** and **2** required for the complete conversion of the initial esters **6eA**, **6pA** did not exceed 0.5 mol % in both cases, while the reaction time was much shorter than in the papers cited above [9,10,11,12,13,14,15,16,17,18,19,20,21,22,23].

After all attempts, the following conditions were found to be optimal for the recyclizations of the allyl derivative (**6eA**): boiling chloroform, 0.5 mol % of **Cat. 1**, 30 min. For synthesis of the methyl substituted **6pA**, the best results were achieved under microwave activation in CH_2_Cl_2_ at 120 °C, 10 min, **Cat. 2** (0.5 mol %).

It was interesting to compare the potential of the commercially available HG-II catalyst with catalysts **Cat.1** and **Cat.2** (see entries 4, 5, 24 and 25 in light gray in Table 4). Analysis of the isolated yields of products **7e**,**p** allows us to conclude that the most popular commercial catalyst (HG-II) is less efficient.

The optimized protocols discussed above were planned to be extended to the recyclization of the remaining esters **6** (Table 2). However, the obvious disadvantage of the method is the need to separate mixtures of diastereomers **6A/6B**. We tried to get around this obstacle, taking into account that *cis*-isomers **6aB**–**6pB** can be considered as undesirable ballast in the target tricycles **7** synthesis, and that the proposed pathway is based on readily available starting materials and reliable synthetic procedures, which allows the synthesis to be scaled up to kilogram quantities. Thereby, the developed approaches were further applied to the metathesis of mixtures of isomers **6A/6B** (Scheme 5). To our satisfaction, the admixtures of *cis*-isomers **6B** had no significant impact on the reaction progress; the polymerization products formed from them could be easily separated during purification by flash or column chromatography to provide the pure cyclopenta[*b*]furo[2,3-*c*]pyrroles **7** in good yields (based on the *trans*-isomer **6A**). Thus, the majority of the RRM reactions presented in Table 2 were carried out using mixtures of isomers **6A/6B** as starting materials.

It worth mentioning here that by revising the ethylene-supported transformations of the similar substrates [9,10,11,12,13,14,15,16,17,18,19,20,21,22,23], the question of the sequence of metathesis stages arose. In theory, two paths exist for the rearrangement of **6** to **7** in an ethylene atmosphere: ROM/RCM, or vice versa, RCM/ROM successions [43]. We suppose that the plausible mechanism of the metathesis of *trans*-isomers **6A** does not differ from the generally accepted one, as it is represented in Scheme 7.

We did not set ourselves the goal of investigating in detail the reaction mechanism; however, from general considerations, it can be assumed that the ring-closing metathesis (RCM) stage has to precede the opening of the 7-oxabicycloheptene cycle (ROM). This statement is supported by the observed behavior of *trans*-3-allyl derivatives **6aA**–**6hA**, which regroup more quickly and under more mild conditions compared to their 3-methallyl analogs **6iA**–**6pA**. This fact is probably related to the more rapid cycloaddition of the double bond of the allylic fragment to the Ru=CH-R center of the catalyst with the formation of structure **I_1_** (Scheme 7). This statement is in good accordance with recently published data [43,44].

During the metathesis of *cis*-isomers **6B**, which are not capable of RCM, the polymer similar to **D** is apparently formed through the cross-metathesis intermediate **C** (Scheme 8). Although the *exo*-cyclic allylic double bond is more reactive, it is also possible to obtain the short-lived triene **8a** which is generated in the course of the catalytic cleavage of the *endo*-cyclic double bond in the starting molecules **6B** (ruthenium complexes **E** and **F**).

## 3. Materials and Methods

### 3.1. General Remarks

All reagents were purchased from commercial suppliers (Acros Organics, Morris Plains, NJ, USA and Merck KGaA, Darmstadt, Germany) and used without further purification. Metathesis reactions required solvents (CH_2_Cl_2_ and CHCl_3_) pre-dried over anhydrous P_2_O_5_ and an inert atmosphere (dry Ar). Thin layer chromatography was carried out on aluminum backed silica pre-coated plates «Sorbfil» and «Alugram». The plates were visualized using a water solution of KMnO_4_ or UV (254 nm). After extraction, organic layers were dried over anhydrous MgSO_4_. IR spectra were obtained in KBr pellets or in thin films using an Infralum FT-801 IR-Fourier. NMR spectra were run in deuterated solvents (>99.5 atom % D) on Jeol JNM-ECA 600 (600.1 MHz for ^1^H and 150.9 MHz for ^13^C) spectrometer for 2–5% solutions in CDCl_3_ or DMSO-d6 at 22–23 °C using residual solvent signals (7.26/77.0 ppm for ^1^H/^13^C in CDCl_3_ and 2.50/39.5 ppm for ^1^H/^13^C in DMSO-*d*_6_) or TMS as an internal standard. CFCl_3_ was used as an internal standard for ^19^F-NMR spectra.

### 3.2. Experimental Procedures

The initial homoallylamines **4a**–**p** were synthesized according to the previously described procedures [45,46]. Carboxylic acids **5a**−**p** were synthesized using the known methods [29,30,31,32]. The detailed description of the preparation for all new compounds obtained in this work is given in the ESI section.

## 4. Conclusions

Our systematic studies on furan chemistry and metathesis reactions resulted in the development of general, effective method for the synthesis of natural product-like cyclopenta[*b*]furo[2,3-*c*]pyrroles from easily available, renewable furan-based precursors. The target tricyclic molecules were formed in high yields from esters of 3-allyl or 3-methallyl substituted 3a,6-epoxyisondolo-7-carboxylic acids in the presence of a new type of Hoveyda–Grubbs catalysts. The influence of the length of the N→Ru coordinate bond in the ruthenium complexes on the rate of the reaction was examined, and selective catalysts for the RRM of the allyl and methallyl derivatives were discovered. It was also proven that the recyclization is strictly stereoselective; only *trans*-isomer from two possible diastereoisomers of the initial methyl 3a,6-epoxyisondolo-7-carboxylates can be involved in the metathesis reaction.

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
