# Peer review of "Application of New Efficient Hoveyda–Grubbs Catalysts Comprising an N→Ru Coordinate Bond in a Six-Membered Ring for the Synthesis of Natural Product-Like Cyclopenta[b]furo[2,3-c]pyrroles"

_molecules, 2020, doi:10.3390/molecules25225379_

Round 1

Reviewer 1 Report

The authors employ their previously reported Hoveyda-Grubbs catalysts (Refs. 27 and 28) for a ring rearrangement metathesis of diastereomeric mixtures of allylated epoxyisoindoles, driving to tricyclic cyclopentafuropyrroles. The synthesis of the starting materials concerning trans/cis stereoselectivity is also explored.

Although these systems are quite specific and thus the work have not too broad applicability, it is certain that few reports can be found about reactions with these natural products-related systems. In addition, the obtained results are better than using other known catalysts. Therefore, this study results interesting enough to justify its publication.

However, some corrections/clarifications should be made, as follows:

1) Line 128. Table 1?

2) Some of the values presented in Table 2 are not too clearly explained. For instance, in entry a of the trans-5A/cis-5B ratio column, the ratio is 67/33. It seems that this should be the ratio presented also in Table 1. However, this ratio corresponds to compound 5i (Table 1) with R1=Me and R2=Ph and not R1=H, R2=Ph (Table 2, entry a). The same happens with other ratios. This should be checked of should be better explained in the text.

3) In Table 2, there is a column with the yield of tricycle 7. In the footnote c and d, no catalyst is mentioned. The values of this column should be mentioned in the text after citing Scheme 5, as the point is missed after all the long trans-cis NOESY and HSQC determinations.

4) Line 236: Scheme

5) Line 294: flash

Author Response

We greatly appreciate the reviewer 1 for the careful reading and complimentary, asserted and valuable comments and suggestions. All changes on the manuscript are yellow highlighted

1) Line 128. Table 1?

 - It should be Table 2. Uncorrected.

2) Some of the values presented in Table 2 are not too clearly explained. For instance, in entry a of the trans-5A/cis-5B ratio column, the ratio is 67/33. It seems that this should be the ratio presented also in Table 1. However, this ratio corresponds to compound 5i (Table 1) with R1=Me and R2=Ph and not R1=H, R2=Ph (Table 2, entry a). The same happens with other ratios. This should be checked of should be better explained in the text.

- It's an editing error. Ratio for all five compounds 5a, 5b, 5g, 5i and 5m were corrected according to data in Table 1.

3) In Table 2, there is a column with the yield of tricycle 7. In the footnote c and d, no catalyst is mentioned. The values of this column should be mentioned in the text after citing Scheme 5, as the point is missed after all the long trans-cis NOESY and HSQC determinations.

- Added catalyst information in the footnote of Table 2. The values of the last column of Table 2 are mentioned in the third part of the manuscript in logical order.

4) Line 236: Scheme

- Corrected

5) Line 294: flash

- Corrected

Sincerely,

Dr. Fedor I. Zubkov, PhD

Reviewer 2 Report

This manuscript describes the full details of the tandem ring-opening and ring-closing olefin metathesis.  Although the same transformation was reported as Ref. 24, significant improvement was achieved by using nitrogen coordinated Ru catalysts.

Chemistries described in this paper would be a nice piece of work and of interest for a number of researchers in this field.

I recommend this manuscript for publication in Molecules.

A minor point:

Page 10, line 2: (Table 2) should be (Table 4).

Author Response

We greatly appreciate the second reviewer for the careful reading and complimentary, asserted and valuable comments and suggestions. All changes on the manuscript are yellow highlighted.

A minor point:

Page 10, line 2: (Table 2) should be (Table 4).

Answer: It was completed clarifying the doubt as (the last column of Table 2 and Table 4).

Sincerely,

Dr. Fedor I. Zubkov, PhD

Reviewer 3 Report

The authors have presented a well-written, comprehensive study of the synthesis of natural product-like cyclopenta[b]furo[2,3-c]pyrroles using efficient Hoveyda–Grubbs catalysts. The optimization investigations undertaken are outstanding examples of how to determine the best conditions for the reaction processes. The NMR and x-ray structural investigations to show configuration and connectivity are well conceived and support the structural and stereochemical assignments.

All compounds are well characterized and indicative of reported compound purity and identity.

Author Response

We greatly appreciate the reviewer for the careful reading and complimentary, asserted.

Dr. Fedor I. Zubkov, PhD